# Hoping for a Better Future during COVID-19: How Migration Plans Are Protective of Depressive Symptoms for Haitian Migrants Living in Chile

**DOI:** 10.3390/ijerph19169977

**Published:** 2022-08-12

**Authors:** Yijing Chen, Claudia Rafful, Mercedes Mercado, Lindsey Carte, Sonia Morales-Miranda, Judeline Cheristil, Teresita Rocha-Jiménez

**Affiliations:** 1Department of Sociomedical Sciences, Mailman School of Public Health, Columbia University, 722 W 168th St, New York, NY 10032, USA; 2Faculty of Psychology, Universidad Nacional Autonoma de Mexico, Circuito Ciudad Universitaria Avenida, C.U., Mexico City 04510, Mexico; 3Center for Global Mental Health, National Institute of Psychiatry, Calz México-Xochimilco 101, Colonia, Huipulco, Tlalpan, Mexico City 14370, Mexico; 4Faculty of Psychology, Universidad Diego Portales, Santiago 8320000, Chile; 5Núcleo de Ciencias Sociales y Humanidades, Universidad de la Frontera, Francisco Salazar 1450, Temuco 4811230, Chile; 6Consorcio de Investigación sobre VIH SIDA TB CISIDAT, Dwight W. Morrow 8, Apt. 7, Centro, Cuernavaca 62000, Mexico; 7Project “When Reality Overcomes the Intention”, Las Condes 7560908, Chile; 8Society and Health Research Center, Facultad de Ciencias Sociales y Artes, Universidad Mayor, Las Condes 4780000, Chile; 9Millennium Nucleus on Sociomedicine (SocioMed), Las Condes 7560908, Chile

**Keywords:** migration, mental and health illness, Chile, COVID-19, social support, racism

## Abstract

This paper explores the migration experiences, perceived COVID-19 impacts, and depression symptoms among Haitian migrants living in Santiago, Chile. Ninety-five participants from eight neighborhoods with a high density of Haitian migrants were recruited. Descriptive statistics, univariate analysis, and logistic regression analysis were conducted. Chi-squared tests were used to confirm univariate results. We found that 22% of participants had major depressive symptoms based on the CESD-R-20 scale, 87% reported major life changes due to COVID-19, and 78% said their migration plans had changed due to the pandemic. Factors associated with more depressive symptoms were being in debt (OR = 3.43) and experiencing discrimination (ORs: 0.60 to 6.19). Factors associated with less odds of depressive symptoms were social support (ORs: 0.06 to 0.25), change in migration plans due to COVID-19 (OR = 0.30), and planning to leave Chile (OR = 0.20). After accounting for relevant factors, planning to leave Chile is significantly predictive of fewer symptoms of depression. Haitian migrants living in Chile had a high prevalence of depression. Planning to leave Chile was a significant protector against depressive symptoms. Future studies should explore how nuanced experiences of uncertainty play out in migrants’ lives, mental well-being, and planning for their future.

## 1. Introduction

Migration has been linked to multiple risk and stress factors that negatively affect the health of migrants, especially their mental health [1,2,3,4,5]. Research has found various pathways and factors that worsen or ameliorate these negative influences, but most of them do not focus on the complexity of the migration process [2,4,6,7,8,9,10]. As the increase in natural disasters, wars, and internal conflicts have placed a growing population of migrants in cities or countries that they did not intend or plan to move to [4,11,12,13,14], few studies have zoomed in on migrants’ trajectories and considered the wide range of possible migration patterns within this population [8,10,15,16,17]. Nevertheless, some studies have found that certain migration trajectories are associated with negative mental health outcomes [2,7,8,10].

During the COVID-19 pandemic, many migrants found themselves in indeterminate circumstances. It is essential to investigate how these migration experiences and patterns impact their mental health status [18,19]. Reports have documented rising levels of anxiety, depression, self-harm, and psychotic symptoms when compared to pre-COVID-19 years [18,20,21]. For some migrant populations, particularly Asians, the limits on movements have posed unique challenges as they could no longer connect with members in their social networks and must cope with rising levels of discrimination [19,20,22]. In general, migrants are often affected by the intersection of class, race, and legal status that hinder their ability to avoid the infection, receive healthcare when needed, and cope with the economic, social, and psychological impacts of the pandemic [23,24,25,26]. Furthermore, COVID-19 has increased precariousness and uncertainty for migrant groups that were already in uncertain situations [13,19,20,23,24,26].

Haitian migrants have a history of migrating to countries within Latin America and the Caribbean, such as Colombia and the Dominican Republic, where they have experienced discrimination, xenophobia, and racism. These have issues translated into social and geographical segregation, and issues regularizing their migration status, and thus Haitian migrants eventually decided to migrate to other countries such as Chile and Brazil [27,28,29]. The first wave of Haitian migrants arrived in Chile in 2015 as this country showed economic and political strength and had relatively welcoming migration policies prior to the 2018 migration reform [30,31]. Furthermore, Chile had an important presence in the United Nations Mission in Haiti (UNMIH) post the 2010 earthquake, which built a relationship between these two countries. In 2015 the Chilean government gave close to 9000 Temporary Resident Visas to Haitians [32]. Nevertheless, the Chilean migration reform in 2018, requiring visa for entry, and other administrative obstacles, countered the ease of entering and staying in Chile for Haitians. Haitians’ necessity to find a safe place to migrate makes them susceptible to the effects of new migration and mobility patterns, given their recent migration to non-traditional destination countries such as Chile and more recently Mexico, as well as complicated landscape of migration policies [33].

In addition, while they are staying in places like Chile and Mexico in uncertain times, the exacerbated socioeconomic conditions and the restrictive immigration policies exacerbated by the pandemic may expose them to greater risks of depression and may change their migration plans [18,23,33].

Given the aforementioned context, this paper aimed to analyze the association between specific migration plans such as remaining in the transit country and potential major depressive symptoms among a sample of Haitian migrants living in Chile. Additionally, in the context of the COVID-19 pandemic and the uncertainty with which Haitian migrants live in Chile, and that hopes and plans for the future may be a protective factor for depression [7], we also hypothesized that those who report planning to stay in Chile would have higher odds of having symptoms of depression in comparison who those who are planning to leave.

## 2. Materials and Methods

### 2.1. Conceptual Framework

Our analysis was guided by a modified version of Bhugra’s Migration Phases and Psychiatric Disorders framework [2]. This framework considers migration experiences or plans while recognizing vulnerabilities (e.g., complex destination migration experiences) and resilience components (e.g., social support) in analyzing the relationship between migration and psychiatric disorders (Figure 1) [2,34,35,36]. Based on previous studies on transit migrants [7,37], the current global migration context, and the restrictive migration policies in Chile, we hypothesized that those who report planning to stay in Chile would have higher odds of having symptoms of depression in comparison with those who are planning to leave. For this analysis, we focused on the post-migration experiences (i.e., their current life in Chile) and their intended or future migration plans.

### 2.2. Data Collection

As a part of an ongoing sequential mixed-method project started in February 2021, we recruited 101 Haitian migrants in Santiago de Chile. The qualitative component of this study aimed to understand more in depth the migration trajectories and mental health status of our sample. For the purposes of this article, we report the quantitative results only. Sampling originally started by inviting participants using the respondent-driven sampling (RDS) method to approximate generalizable samples of segregated populations [38,39,40]. However, as data collection progressed, we observed that the contacts of those who planned to migrate out of Chile were less likely to be invited than the contacts of those planning to stay. Therefore, we decided to switch to a two-stage conglomerate sampling approach. The first stage involved subsampling and stratifying neighborhoods with a higher density of Haitian migrants in the Metropolitan Region of Santiago [41]. The neighborhoods with a higher density of Haitian migrants also have migrant camps inside them where a significant number of Haitian migrants live. We also selected neighborhoods based on the field staff’s community presence, and those were also the safest neighborhoods. The neighborhoods selected (e.g., Cerrillos, Estación Central, Santiago Centro) are mainly located in downtown Santiago and in the south of the city and have a historical presence of migrant camps. In the second stage, residents in each selected neighborhood were invited to participate. No one was excluded, and everyone who met the eligibility criteria and gave consent completed the survey.

Eligibility criteria for this study included (a) 18 years old or older; (b) Spanish or Creole speaker; (c) willing and able to provide informed consent; (d) born in Haiti, and (e) living in Chile at the time of the interview. The instruments used for this study were prepared in Spanish and translated to Creole (i.e., written consent forms, informational flyer, survey). Two members of the field staff were bilingual, so the participant could choose to complete the survey in Spanish or Creole. Staff provided a copy of the consent form and explained the study to potential participants, including the purpose, procedures to be followed, and risks and benefits of participation. Along with the consent form, staff provided a flyer with relevant mental health information, resources, and the project psychologist’s contact information in case any participant wanted to speak with her in private or schedule an appointment. Upon receipt of written consent, staff conducted face-to-face interviews using Survio, a computer-assisted personal interviewing technology (https://www.survio.com/es/, accessed on 15 February 2021). We conducted all the surveys face-to-face in open spaces (i.e., outside participants’ houses, in parks) or, when preferred by the participant, inside their homes or in chapels. The survey took 30–50 min to complete and included questions on sociodemographics, identity, discrimination, social support, migration plans and experiences, perceived consequences of COVID-19, access to health services, geospatial elements (e.g., living and working locations), and a mental health scale to evaluate possible symptoms of depression (i.e., CESD-R-20 items).

### 2.3. Ethics Procedure

We invited representatives from three key local migrant organizations, one of which was founded by Haitian migrants, to form a Community Advisory Board (CAB). We met with each organization via Zoom to discuss the following items: (a) recruitment and the invitation process, (b) the amount and type of compensation, (c) language, length, and content of the survey, and (d) dissemination strategies. The role of the CAB was essential to recognize the power imbalance between the researchers and participants, and the importance of having the community involved in the decision-making processes of the study [42,43]. After meeting with the CAB, we confirmed the importance of not asking for any private information, including names or personal identification numbers (i.e., RUN, used in official documents and purchases in Chile). Each participant had a code, and we only collected contact information if they wanted to meet with the psychologist or if they wanted to participate in the qualitative component of the study. Participants were compensated with a $6 USD gift card to be used in a grocery store (i.e., Lider) for participating in the interview. The project was approved by the Scientific Ethics Committee of the PI University.

### 2.4. Dependent Variable

Depressive symptoms were assessed with the self-reported Center for Epidemiologic Studies Depressive Scale—Revised 20-item (CESD-R-20) [44,45]. The CESD-R-20 defines a score equal to or above 16 (out of 60) at risk for clinical depression. An adapted Spanish version of the CESD-R-20 instrument was used in this study [46,47,48,49]. Bilingual staff translated the CESD-R-20 scale from Spanish to Creole, then we did a back-translation to ensure that the first translation was accurate. Details of the validation process are explained elsewhere [50]. As the participants had the option to choose their preferred language for the survey, some responded to the original CESD-R-20 in Spanish (*n* = 32) and others in Creole (*n* = 63).

### 2.5. Independent Variables

Sociodemographic variables included age, gender, marital status (i.e., married or common-law vs. single, divorced, or widowed), self-defined group membership (e.g., migrant and/or Haitian, and/or Chilean), education level, religious affiliation and service attendance, and perceived financial situation (See Table 1).

#### 2.5.1. Migration Experiences

Participants were given a mix of multiple choice and open-ended questions. Participants’ history of migration was constructed by asking the age in which they left their country of origin, the length of their stay in Chile, reasons for leaving their country of origin, migration plans (e.g., Did you want to migrate to Chile when you left your country?), reasons for wanting to leave Chile (for those who expressed wanting to depart), and whether their migration plans had changed due to the COVID-19 pandemic (Table 2). We also asked questions about any experiences of discrimination while living in Chile.

#### 2.5.2. Social Support

We asked four questions of the 19-item Medical Outcome Study Social Support Survey (mMOS-SS) [51]. We included those that the community and CAB recommended (i.e., have someone to talk to about your anxieties, have someone to help you when you get sick, have someone to lend you money when you need it, and have someone to show you love and affection) and we also asked about that person’s nationality.

### 2.6. Statistical Analysis

Descriptive statistics were calculated to provide an overview of recent migrants’ demographics by the possible status of clinical major depression. A bivariate analysis was performed on discrete variables to calculate the odds ratio of having a CESD-R-20 score of at least 16 given each predictor (Table 2). We also performed Chi-squared tests as a non-parametric method to confirm our odds ratio results. Since all the *p*-values were very similar, we did not include details about them in the table. Bivariate logistic regression was performed to identify factors associated with possible major depressive symptoms. In the event of two highly correlated covariates, the one with the strongest association with the outcome was retained. To ensure the integrity of the model, interactions between predictors and key independent variables such as gender were calculated, but no significant interaction was found. Only variables significant at *p* < 0.05 were retained in the final multivariate model (Table 3). All regression models are presented with crude and adjusted odds ratios (OR), 95% confidence intervals (95%CI), with *p* < 0.05 considered significant. Analyses were conducted using the readr, dplyr, fmsb, and ltm packages in RStudio Version 1.4.1106 [52,53,54,55,56].

## 3. Results

Among the 101 participants surveyed, only 95 were included in the analyses due to incomplete items on the CESD-20 scale. The sociodemographic characteristics for the overall sample and by presence of clinical major depression are provided in Table 1. With regards to self-reported group membership, we found that even though all of the participants were born in Haiti, only 68% self-identified as belonging to the Haitian group. The remaining viewed themselves as migrants (31%) or Chilean (4%). Those who self-reported as a migrant had more odds of having depressive symptoms. Most of the sample reported some level of formal education (87%), having children (71%), and being affiliated with a religion (77%). About 78% of participants reported planning to move to Chile in the first place (i.e., when they were still in Haiti). On average, our participants had stayed in Chile for 4.58 years. Overall, most of our sample (87%) reported having suffered consequences (mostly economic issues) due to the pandemic. The Cronbach’s alpha (α) was calculated to assess the reliability of CESD-R-20 Scale in this study, and values ranged from 0.830–0.927 for the 20 items in the current study sample. The Cronbach’s alpha coefficient for the 20-item-scale was 0.895, indicating a high internal consistency and that the removal of any of the items would have reduced the overall alpha marginally [57,58,59].

Univariate associations between each predictor and symptoms of depression are presented in Table 2. Self-identifying as a migrant (OR = 3.42, 95% CI = 1.25–9.37, *p* = 0.014), having a good self-perceived financial situation (OR = 8.85, 95%CI = 1.31–59.80, *p* = 0.010), and being in debt (OR = 3.43, 95%CI = 1.03–11.45, *p* = 0.039) predicted significantly higher odds of having depressive symptoms, while self-identifying as a Haitian (OR = 0.31, 95% CI = 0.12–0.86, *p* = 0.021) as well as being interviewed in Creole (OR = 0.12, 95% CI = 0.04–0.36, *p* < 0.001) predicted significantly lower odds of scoring as clinically depressed.

Social support and discrimination predictors were significantly associated with odds of having symptoms in the univariate analysis. Participants who were not able to move to another neighborhood had significantly higher odds (OR = 6.19, 95% CI = 1.99–19.23, *p* < 0.001) of having symptoms of depression. All the social support items predicted significantly lower odds of having depression. Identifying with a religion was not a significant predictor (OR = 0.33, 95% CI = 0.09–1.19, *p* = 0.082), but going to services weekly significantly predicted higher odds of presenting clinical depression symptoms (OR = 3.96, 95% CI = 1.03–15.19, *p* = 0.036).

A few predictors examining migration experiences and COVID-19 impacts were found to be significantly associated with lower odds of depressive symptoms: planning to come to Chile when leaving the country of origin (OR = 0.35, 95% CI = 0.12–1.00, *p* = 0.047), planning to leave Chile (OR = 0.20, 95% CI = 0.06–0.62, *p* = 0.003), having migration plans changed due to COVID-19 (OR = 0.30, 95% CI = 0.10–0.88, *p* = 0.024), and deciding to move to a country different than originally planned (OR = 0.11, 95% CI = 0.02–0.50, *p* = 0.001).

The multivariate analysis is presented on Table 3. Although not significant in the bivariate analyses, based on the literature and Bhugra’s conceptual framework, we included sex, self-perceived financial situation, and religious affiliation as covariates in the model. After adjusting for the covariates, planning to leave Chile remained a significant protective factor for depressive symptoms (Adjusted odds ratio (AOR) = 0.17, 95% CI = 0.04–0.68, *p* = 0.0131). The social support item of having someone with whom to talk about anxieties had a significant negative association with symptoms of depression (AOR = 0.06, 95% CI = 0.01–0.28, *p* = 0.0006) whereas the discrimination item of not being able to move to another neighborhood was positively associated with having depressive symptoms (AOR = 3.88, 95% CI = 0.94–16.57, *p* = 0.06). Additionally, we report here the nationality results of support persons in their lives.

## 4. Discussion

The prevalence of symptoms of depression among a sample of Haitian migrants (*n* = 95) living in Chile during COVID-19 was 22%. This is similar to the prevalence of depression in the general Chilean population during the pandemic: 22.6% [60,61]. Although the prevalence is similar to some studies conducted with the general population, such studies were conducted online or by phone which may have influenced the responses and perception of the isolation. Given that previous research found that migrants had a prevalence of depression between 14 and 20%, our sample presented a higher percentage within the migrant population prior to COVID [6,62,63]. As studies found that migrants during COVID had a prevalence of depression between 23 and 28%, our sample of Haitian migrants in Chile had a lower percentage of depression [24,64]. Nevertheless, studies conducted during COVID often used web-based surveys in English, and results could be heavily influenced by migrants’ English proficiency and whether they had access to Internet devices. Furthermore, one cross-sectional study in 2001 using structured psychiatric interviews found that about 5.5% of members of private households in Santiago de Chile had depression [65].

As hypothesized, we found that after controlling for relevant covariates, planning to stay in Chile among our respondents was significantly associated with depressive symptoms. The consequences of the COVID-19 pandemic have been especially harsh on populations that work in the informal sector, including migrants [19,20,22], who, given the strict confinement measures implemented in Chile, had very few employment opportunities [66]. An important percentage of this study’s participants (77%) reported experiencing economic issues during the COVID-19 pandemic and, when we asked if their migration plans had changed due to the COVID-19 pandemic, 84% of those who were planning to leave responded affirmatively. The changing conditions of migration policies, the pandemic, and US–Mexico border conditions have forced migrants to change their migration trajectories. Deciding on their new migration plans could spark new hopes and make migrants more optimistic about their potential futures, and thus having their plans changed due to COVID and changing their migration destination are both protective of their mental health.

The political climate for migrants in Chile, particularly for Haitian migrants, has been especially difficult as well. The migration reform in 2018 established a consular visa requirement for entry into Chile which had restrictive effects on the arrival of Haitian migrants. In addition, there were administrative obstacles (e.g., the request for a criminal record certificate be sent to Chile from Haiti) which increased monetary costs and waiting times for the completion of regularization procedures and limited the possibilities of regularizing the migratory situation for Haitian migrants already in Chile. These situations had particularly serious effects on the ability of Haitian migrants to obtain and/or maintain a job, increasing the financial difficulties for this group [67]. Simultaneously, the 2020 election of a new president in the United States renewed hopes that Haitian migrants would have the opportunity to enter that country. The hardships faced in Chile coupled with the hope and actual entrance of some migrants from Haiti to the United States drove the Haitian migration flow from Chile to the Mexico–US border [68]. Both the United States and Mexico witnessed an increase in Haitian nationals in case and application numbers [69,70].

In the present study, a high percentage of migrants rated their financial situation as bad or very bad (73%). This is reflective of the tough conditions Haitians are experiencing in Chile as a result of the factors discussed above, many of which were exacerbated by COVID-19. The unemployment rate was high (60%), consistent with reports citing difficulty finding jobs [21,23]. Surprisingly, being in a better self-rated economic situation did not protect individuals from depressive symptoms; rather, it was a risk factor for depression. This might be due to how migrants of poorer financial standings were forced to leave Chile, and planning for a potentially better future gives them hope. On the other hand, those who have better financial standings need to face many obstacles to stay in Chile, putting a heavier mental drain on them. Nevertheless, more research is needed to investigate the interaction effects of economic status and migration plans on mental health [5,25,71,72].

Our results confirmed previous research on the protective effects of social support on mental health. Items from financial to emotional social support were all associated with fewer depressive symptoms, confirming the important role of social support discussed in the literature [4,6,35,73,74,75,76,77]. Our data supported the direct protective effects of social support, measured as the availability of somebody to help, against depressive symptoms. In the multivariate analysis, the social support factor of having someone to talk to about concerns came out as the most influential factor in the social support block. Hence, the present results suggested that having social support for concerns was most impactful against symptoms of depression. However, our sample gave an interesting result on religion. On one hand, identifying with a religion was protective against depression, consistent with previous studies that found religiosity a protective factor of mental health [78,79]. On the other hand, going to services weekly was found to be a risk factor for depression among those who are religious. This might be because migrants who go to services weekly search for help in religion regularly instead of engaging in practical solutions, having a higher odds for depression. Hence, those who identify with a religion but do not go to services weekly enjoy the benefits of religion but do not overly rely on religion for solving their issues. Future studies should examine how various forms of social support may have differential impacts on different aspects of mental health and physical health.

Discrimination is a key risk factor for migrants’ mental health, especially during COVID-19. As a moving population, migrants may experience heightened levels of discrimination and stigma and may have worsened mental well-being from these experiences. Racial and ethnic discrimination is significantly associated with worse mental health status [6]. A lower length of residence and higher acculturation level are associated with stronger effects of discrimination [80,81]. According to the WHO Apart Together survey, around 40% of those living on the streets or in insecure accommodation reported discrimination, with the unemployed reporting even higher levels of discrimination [21]. An important part of our sample (68%) identifies themselves as migrants, and such a perception had significantly higher odds of having symptoms of depression, which may be illustrating the perception of not being part of the destination community and self-perceiving the sense of otherness. This has been widely explored and has found to be associated with mental health issues among migrant communities throughout the world [82,83]. We plan to explore the relationship between self-identifying as a migrant and depression in a future paper related to the current project.

Contrary to our expectation that multiple factors of discrimination would be predictive of higher depressive symptoms, only one factor—being unable to move to another place—was significantly associated with depression status. Previous studies on (im)mobility among migrants demonstrated how structural factors such as discrimination and socioeconomic inequality can produce immobility and constitute the sense of insecurity [84,85]. A large portion of our sample also reported coming to Chile as a temporary destination, with the hope of migrating to other countries such as the United States in the future—and the recent situation had pushed some of them to make the decision to leave Chile. Furthermore, given that Chile is not the final destination for a large portion of our sample, it is plausible that the inability to move was especially concerning for this group. This trend is also consistent with studies in other contexts in which migrants who freely chose to move had a lower prevalence of depression [24,62].

Besides the diversity of migrant characteristics, migration encompasses a broad range of experiences and trajectories on spatial and temporal scales that cannot be captured with the migrant versus non-migrant categorization [8,10]. Migration is dynamic and multi-directional: as migrants move from place to place, their plans and resources may change such that they would change the duration of stay, or they would end up in a different place than their original destination [8,13]. These nuances are integral parts of the migration experience and present a multitude of possibilities under the term migration, with risk and protective factors at each step. Bhugra outlined the vulnerability and resilience factors in the process of migration during premigration, selection for migration, the migration experience, and post-migration [2]. At each of the four stages, the migrant might experience a broad range of factors that may either facilitate better mental health or lead to worsened mental health.

The present investigation focused on the condition of transit migration where Haitians came to Chile with the plans of continuing migration in the future. As this migrant collective faced the challenges of policy changes, economic distress, and COVID-19, they could build resilience against mental health issues by having hope for a better future and planning for a journey to achieve that. While conducting this study, we heard from many participants that a family member or friend had been able to cross to the United States, a fact that has been documented in the press [33,68,86]. Thus, this could be another reason for finding a protective effect between the planning to leave Chile and symptoms of depression. Other studies have found a similar effect when plans, resilience, and the future have been analyzed and found to be protective factors for migrants’ mental health [7,16,87]. Optimism and a sense of control were protective mediators for migrants’ mental health while a lack of hopefulness about the future was associated with worse effects on mental health [5,87].

## 5. Conclusions

This study surveyed Haitian migrants living in Santiago, Chile about their migration experiences and major depression status. Haitian migrants experienced a myriad of vulnerabilities and uncertainties due to the complex policy environment during the COVID-19 pandemic in Chile. The data revealed that Haitian migrants had a higher level of prevalence of major depression than the general public in Chile, consistent with our hypotheses and previous literature on migrants’ mental health during COVID-19. We found that being in debt and discrimination were risk factors associated with the presence of major depression. Protective factors against clinical depression included social support, change in migration plans due to COVID-19, and planning to leave Chile. Multivariate analysis highlighted that planning to leave Chile is significantly predictive of the absence of major depression. These results suggest that Haitian migrants in Chile suffer from a higher prevalence of major depression and present risk and protective factors associated with migrants’ experiences in the country.

Although this study showed the important effects of various factors during the migrant’s journey on their mental health status and there are no other studies assessing the prevalence of symptoms of depression on Haitian migrants, we cannot generalize the experiences of all Haitians living in Chile as this is a cross-sectional study and includes a small sample. Moreover, our sample size limited the accuracy of our estimates as demonstrated by wide confidence intervals in our analyses. Our study does illustrate an interesting phase of Haitian migrants’ journey that may be useful for informing practice of community advocacy groups and health services providers, as well as future studies conducted in Chile but also throughout the Americas as Haitians are migrating northward [86]. Future research should delve into the various parts of the migration process following Bhugra’s Migration Phrases and Psychiatric Disorders Framework [2] and investigate how the same factors such as social support and sense of control may have differential impacts on these stages of migration. Longitudinal cohort designs or studies including other destinations of the same migrant group can reveal the nuances of these impacts over time and possibly expose valuable windows for intervention. Studies can also probe the relationship between optimism, sense of control, and other psychiatric disorders.

## Figures and Tables

**Figure 1 ijerph-19-09977-f001:**
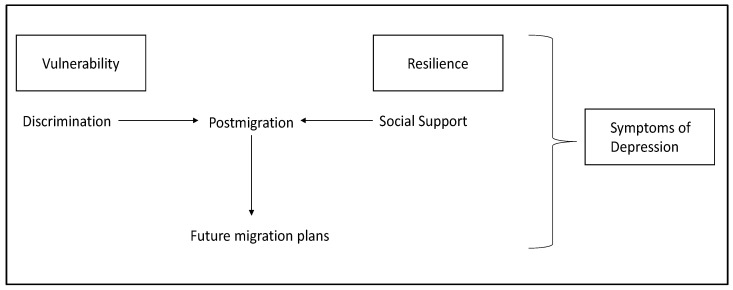
Migration Experiences and Psychiatric Disorders Framework.

**Table 1 ijerph-19-09977-t001:** Sociodemographic characteristics by symptoms of depression among Haitian migrants in Santiago de Chile 2021 (*n* = 95).

Characteristics	Total Sample (*n* = 95)	No Symptoms of Depression (*n* = 74) 78%	Symptoms of Depression (*n* = 21) 22%
Age [median (IQR)]	33 (28–37)	31 (27–37)	34 (29–39)
Gender			
Male	47 (49%)	39 (53%)	9 (38%)
Female	48 (51%)	35 (47%)	13 (62%)
Self-reported group membership ^1^			
Migrant	29 (31%)	18 (24%)	11 (52%)
Haitian	65 (68%)	55 (74%)	10 (48%)
Chilean	4 (4%)	3 (4%)	1 (4%)
Marital status			
Married/with partner	46 (48%)	33 (45%)	13 (62%)
Single/no partner	49 (52%)	41 (55%)	9 (38%)
Level of education			
Less than elementary school	12 (13%)	9 (12%)	3 (14%)
Completed elementary school	42 (44%)	31 (42%)	11 (52%)
Completed secondary school	38 (40%)	32 (43%)	6 (29%)
Completed university	3 (3%)	2 (3%)	1 (4%)
Has children	67 (71%)	50 (68%)	17 (81%)
Religion			
Affiliated with a religion	73 (77%)	67 (91%)	16 (76%)
Planned to move to Chile in the first place	74 (78%)	61 (82%)	13 (62%)
Length of stay in Chile in years [median (IQR)] (*n* = 86)	4.58 (4–5)	4.59 (4–5)	4.53 (4–5)

^1^ These groups are not mutually exclusive.

**Table 2 ijerph-19-09977-t002:** Univariate association with symptoms of depression among Haitian migrants in Santiago de Chile (*n* = 95).

Predictor	Count [CESD < 16] *n* = 74	Count [CESD ≥ 16] *n* = 21	Odds Ratio (95% CI)	*p*-Value
*Sociodemographic*				
Gender (female)	35 (47%)	13 (62%)	1.81 (0.67, 4.88)	0.240
Age	32.51	33.52	-	-
Marital status (married)	33 (45%)	13 (62%)	2.02 (0.75, 5.45)	0.164
Self-reported group membership				
Migrant	18 (24%)	11 (52%)	3.42 (1.25, 9.37)	0.014 *
Haitian	55 (74%)	10 (48%)	0.31 (0.12, 0.86)	0.021 *
Level of education (did not complete primary school)	9 (12%)	3 (14%)	1.20 (0.29, 4.92)	0.797
Have kids	50 (68%)	17 (81%)	2.04 (0.62, 6.73)	0.238
Have children living in Haiti	28 (39%)	5 (24%)	0.51 (0.17, 1.56)	0.236
Language of interview (Creole)	57 (77%)	6 (30%)	0.12 (0.04, 0.36)	<0.001 **
*Financial Situation*				
Unemployed	48 (65%)	12 (57%)	0.72 (0.27, 1.94)	0.520
Self-rated financial situation				
Very good or good	2 (3%)	3 (14%)	8.85 (1.31, 59.80)	0.010 *
More or less	13 (18%)	8 (38%)	3.63 (1.20, 10.98)	0.019 *
Bad or very bad (reference)	59 (79%)	10 (48%)	-	-
In debt (*n* = 92)	8 (11%)	6 (30%)	3.43 (1.03, 11.45)	0.039 *
*Social Support and Discrimination*				
Discrimination				
Treated with respect (*n* = 93)	49 (81%)	11 (55%)	0.60 (0.22, 1.64)	0.318
Insulted or offended by people	36 (49%)	11 (52%)	1.16 (0.44, 3.06)	0.764
Treated the same way than others in stores	50 (68%)	15 (71%)	1.20 (0.41, 3.48)	0.738
Unjustly fired from jobs (*n* = 94)	10 (14%)	5 (24%)	1.97 (0.59, 6.57)	0.267
Not able to move to another neighborhood	12 (16%)	9 (43%)	6.19 (1.99, 19.23)	<0.001 ***
Social support: have someone who can… (nationality of that person)				
Talk about anxieties (82 Haitian, 1 Chilean)	71 (96%)	12 (57%)	0.06 (0.01, 0.24)	<0.001 ***
Help when sick (85 Haitian, 1 Chilean)	71 (96%)	15 (71%)	0.11 (0.02, 0.47)	<0.001 ***
Lend money (69 Haitian, 2 Chilean, 1 Other)	62 (84%)	10 (48%)	0.18 (0.06, 0.51)	<0.001 ***
Show love and affection (88 Haitian, 1 Haitian)	71 (96%)	18 (86%)	0.25 (0.05, 1.36)	0.091
Identify with a religion	67 (91%)	16 (76%)	0.33 (0.09, 1.19)	0.082
Go to services weekly (*n* = 83)	35 (52%)	13 (81%)	3.96 (1.03, 15.19)	0.036 *
*Migration Experiences*				
Age at leaving the country of origin (mean)	26.11	26.19	-	-
Reasons for leaving country of origin				
Seek dreams and life changes	49 (66%)	11 (52%)	0.56 (0.21, 1.50)	0.249
My family or partner migrated	6 (8%)	1 (5%)	0.57 (0.06, 4.99)	0.606
No work there	6 (8%)	1 (5%)	0.57 (0.06, 4.99)	0.606
Planned to come to Chile when leaving the country of origin	61 (82%)	13 (62%)	0.35 (0.12, 1.00)	0.047 *
Time in Chile	4.54	4.53	-s	-
Planning to leave Chile	66 (89%)	13 (62%)	0.20 (0.06, 0.62)	0.003 **
Reasons for planning to leave Chile (*n* = 79)				
Work opportunities	1 (2%)	5 (38%)		
Discrimination/racism	36 (55%)	5 (38%)	0.52 (0.15, 1.76)	0.292
Cannot bring family here	5 (6%)	0 (0%)		
*COVID-19-Related*				
Had suffered consequences due to the pandemic	64 (86%)	19 (90%)	1.48 (0.30, 7.37)	0.629
Economic issues	57 (77%)	15 (71%)	0.75 (0.25, 2.22)	0.599
Change of living place	7 (9%)	1 (5%)	0.48 (0.06, 4.12)	0.496
Anxiety and worry	12 (16%)	4 (19%)	1.22 (0.35, 4.25)	0.761
Migration plans changed due to COVID-19 (*n* = 93)	61 (84%)	12 (60%)	0.30 (0.10, 0.88)	0.024 *
Move to another country (*n* = 70)	54 (92%)	6 (55%)	0.11 (0.02, 0.50)	0.001 **
Move back to Haiti (*n* = 70)	3 (5%)	1 (9%)	1.87 (0.18, 19.80)	0.602

*: *p* < 0.05, **: *p* < 0.01, ***: *p* < 0.001.

**Table 3 ijerph-19-09977-t003:** Multivariate Logistic Regression of Planning to Stay in Chile and Symptoms of Depression among Haitians in Santiago de Chile (*n* = 95).

Coefficients	Estimate	Std. Error	z Value	*p* Value	OR (95%CI)
(Intercept)	2.3420	1.6937	1.383	0.1667	10.4021 (0.3898, 331.1515)
Planning to leave Chile	−1.7752	0.7154	−2.482	0.0131 *	0.1694 (0.0394, 0.6772)
Have someone to talk about anxieties (social support)	−2.8385	0.8396	−3.381	0.0007 ***	0.0585 (0.0096, 0.2768)
Not able to move to another neighborhood (discrimination)	1.3570	0.7217	1.880	0.0601	3.8844 (0.9388, 16.5659)
Having a religious affiliation	−1.0440	0.8344	−1.251	0.2108	0.3521 (0.0681, 1.8933)
Being female	0.3320	0.6749	0.492	0.6227	1.3938 (0.3643, 5.4273)
Being in a bad financial situation	0.9933	1.1794	0.842	0.3996	2.7003 (0.2607, 31.2644)

*: *p* < 0.05, ***: *p* < 0.001.

## Data Availability

The data presented in this study are available on request from the corresponding author. The data are not publicly available due to ethics and confidentiality concerns for the participants.

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
