# Peer review of "Hoping for a Better Future during COVID-19: How Migration Plans Are Protective of Depressive Symptoms for Haitian Migrants Living in Chile"

_ijerph, 2022, doi:10.3390/ijerph19169977_

Round 1

Reviewer 1 Report

Dear authors

Thank you this interesting and novel study. I think it deserves attention and dissemination, yet it could be improved if some precisions and clarifications are considered in a revised version of the manuscript:

- Abstract: add the criterion for selection of neighbourhoods

- Abstract and methods: clarify whether non parametric tests were considered given small sample size.

- Abstract: some ORs are unclear or difficult to understand: discrimination (ORs=0.60-6.19); social support (ORs=0.06-0.25).

- Intro: a bit more about the historical lens of Haitian migrants in LAC, please, connecting with racism, social exclusion and xenophobia.

- Intro: please consider expanding more the migration flows of Haitian migrants in the region in past decades and why/how they started to arrive to Chile...

- Methods: please name the boroughs that were selected for participants recruitment, and explain where they are located/general description in the metropolitan area. This could help the reader understand better the context in which participants responded the survey.

- Discussion: please explain main findings in the context of Haitians migration flows... did they wanted to leave Chile before the pandemic kicked-in? how do you explain in the migration of this particular community process-context that wanting to leave was protective?

- Dicusssion: it is quite interesting that a group of them did not define as Haitians, could you explain this finding a bit more. As explained by authors: "not everyone (only 68%) identified themselves as belonging to the 193 Haitian group"

- Discussion: could you expand a bit more the findings around discrimination in this particular group? How does it interact with any measure of social integration you included in this study? (different to social support which might come from their own communities... I refer to intergroup interactions)

- Discussion and conclusion: sample size is a true limitation of this study, could you please discuss it more and justify the uniqueness of this study regardless of this limitation (internal consistency of participants with such particular profile: low-income, marginalised Haitian migrants living in poor quality neghbourhoods/ transient camps, in selected boroughs in the metropolitan area of Chile?)... How is this novel to Chile and the region? 

- Discussion and conclusion: please consider adding the study´s implications for policy and practice?

Thanks and congratulations on this very interesting paper.

Author Response

Dear Guest Editor

Dr. Ernesto Castañeda and Reviewers,

We appreciate all the comments and suggestions made to our manuscript entitled “Hoping for a better future during COVID-19: How migration plans are protective of depressive symptoms for Haitian migrants living in Chile” (1841747). We have responded to each comment and suggestion and have addressed them in the main document. See our responses below:

Sincerely,

- Abstract: add the criterion for selection of neighborhoods

We have specified in the abstract that the criterion for the selection of the neighborhood was based on the high density of migrants living in such neighborhoods. The abstract now reads like this:

This paper explores the migration experiences, perceived COVID-19 impacts, and depression symptoms among Haitian migrants living in Santiago, Chile. 95 participants from 8 neighborhoods, with a high density of Haitian migrants, were recruited.

- Abstract and methods: clarify whether non parametric tests were considered given small sample size.

We added details in the abstract and methods about chi-squared test results and how they corroborated our univariate analyses:

Descriptive statistics, univariate analysis, and logistic regression analysis were conducted. Chi-square tests were used to confirm univariate results.

We also performed chi-squared tests as a non-parametric method to confirm our odds ratio results. Since all the p-values were very similar, we did not include details on them in the table.

- Abstract: some ORs are unclear or difficult to understand: discrimination (ORs=0.60-6.19); social support (ORs=0.06-0.25).

We changed our wording in the abstract to clarify this.

Factors associated with more depressive symptoms are being in debt (OR=3.43) and discrimination (OR range from 0.60 to 6.19). Factors associated with less odds of depressive symptoms were social support (OR range from 0.06 to 0.25), change in migration plans due to COVID-19 (OR=0.30), and planning to leave Chile (OR=0.20).

- Intro: a bit more about the historical lens of Haitian migrants in LAC, please, connecting with racism, social exclusion and xenophobia.

We appreciate this comment and we understand how important is to incorporate the historical perspective of Haitian migrants in Latin America and the Caribbean and how experiences of racism, social exclusion and xenophobia play an important role in this migrant group. Therefore, we have added in the introduction a few sentences addressing such phenomenon:

Haitian migrants have a history of migrating to countries within Latin America such as Dominican Republic where they have experienced discrimination, xenophobia and racism. These issues translate into social and geographical segregation, issues regularizing their migration status, and thus Haitian migrants eventually decided to migrate to other countries such as Chile and Brazil [27-29].

- Intro: please consider expanding more the migration flows of Haitian migrants in the region in past decades and why/how they started to arrive to Chile.

We have added a paragraph explaining the experiences of racism of this migrant group but also we included more context on why the Haitian community started moving to Chile and the importance of this migrant group in Chile:

Haitian migrants have a history of migrating to countries within Latin America such as Dominican Republic where they have experienced discrimination, xenophobia and racism. These issues translate into social and geographical segregation, issues regularizing their migration status, and thus Haitian migrants eventually decided to migrate to other countries such as Chile and Brazil [27-29]. The first wave of Haitian migrants arrived to Chile in 2015 as this country showed economic and political strength and had relatively welcoming migration policies prior to the 2018 migration reform [30,31]. Furthermore, Chile had an important presence in the United Nations Mission in Hati (UNMIH) post the 2010 earthquake which built a relationship between these two countries. In 2015 the Chilean government gave close to 9,000 Temporary Resident Visas to Haitians [32].

- Methods: please name the boroughs that were selected for participants recruitment, and explain where they are located/general description in the metropolitan area. This could help the reader understand better the context in which participants responded the survey.

Thank you for this suggestion we have added a few more details regarding the location where participants were invited to participate. We have added some more details of the location and the characteristics of such neighborhoods:

The neighborhoods selected (e.g., Cerrillos, Estación Central, Santiago Centro) are mainly located in downtown Santiago and in the South of the city and have a historical presence of migrant camps.

- Discussion: please explain main findings in the context of Haitians migration flows... did they wanted to leave Chile before the pandemic kicked-in? how do you explain in the migration of this particular community process-context that wanting to leave was protective?

We appreciate this question, and we agree that more reflection is needed. Although we do not know if they had plans of moving before the pandemic, we did ask if their migration plans changed due to COVID-19 and 84% of our sample said yes; therefore, we believe than it is something that definitely impacted their decision especially as many of them were impacted by the pandemic, financially (77%) or otherwise. The Haitian migrant group’s condition was already precarious before the pandemic but some of the consequences of it triggered the idea of leaving Chile. We have added a few sentences highlighting that not only hoping and planning to leave and find an opportunity in the United States but also having family members or friends crossing to the United States may have enhanced this hope and prevented them to show depressive symptoms as some of them believe and planned their trip based on this hope:

While conducting this study we heard from many participants that a family member or friends had been able to cross to the United States, a fact that has been documented in the press [31,62,79]. Thus, this could be another reason of finding a protective effect between the planning to leave Chile and symptoms of depression.

- Discussion: it is quite interesting that a group of them did not define as Haitians, could you explain this finding a bit more. As explained by authors: "not everyone (only 68%) identified themselves as belonging to the Haitian group"

This is a great question, and we have clarified that participants had the choice of defining themselves as migrants, Haitians, and Chileans and we have clarified that those who self-perceived as migrants had higher odds of showing symptoms of depression:

In regard to self-reported group membership, we found that even though all of the participants were born in Haiti, not everyone (only 68%) identified themselves as belonging to the Haitian group, the others self-perceived themselves as migrants (31%) and only 4% as Chilean. Those who self-reported as a migrant, had more odds of having depressive symptoms.

- Discussion: could you expand a bit more the findings around discrimination in this particular group? How does it interact with any measure of social integration you included in this study? (different to social support which might come from their own communities... I refer to intergroup interactions).

This is a very interesting question and we did include questions surrounding discrimination and social integration as well as social networks but given that is a complex and extensive theme, we are writing another manuscript solely analyzing how discrimination experiences may be associated with symptoms of depression. However, we have added a few sentences in the discussion clarifying that the fact of self-identifying as migrants may be illustrating further experiences of discrimination and having the sense of otherness may be explaining such association between feeling migrant and symptoms of depression in our bivariate analysis (Table 2):

An important part of our sample (68%) identifies themselves as migrants, and such perception had significantly higher odds of having symptoms of depression, which may be illustrating the perception of not being part of the destination community and self-perceiving the sense of otherness. This has been widely explored and has found to be associated with mental health issues among migrant community throughout the world [76,77]. We plan to explore the relationship between self-identifying as a migrant and depression in another manuscript under the project.

- Discussion and conclusion: sample size is a true limitation of this study, could you please discuss it more and justify the uniqueness of this study regardless of this limitation (internal consistency of participants with such particular profile: low-income, marginalised Haitian migrants living in poor quality neghbourhoods/ transient camps, in selected boroughs in the metropolitan area of Chile?)... How is this novel to Chile and the region? 

We understand that the sample is a true limitation, but we believe that this is a unique ongoing study for several reasons: We captured migrants in an exceptional moment that includes the COVID-19 pandemic and in a new migration phase, as some of them were planning to emigrate and others were doing it as we were collecting this data. We hope that these findings are useful to understand Haitian migrants’ mental status as the migrate thorough the continent up to the Mexico-United States border and provide some input for the community, health services providers, and researchers. Additionally, there are no studies that assess symptoms of depression among this migrant collective in Chile despite that they have been in this country at least for 5 years. Although we only focused on Santiago, we included participants of 8 different neighborhoods to try to capture the diversity within this migrant community so not everyone sampled was living in marginalized camps. We have added a few more details in the conclusion to address such limitations but also to highlight this study’s contributions:

Although this study showed the important effects of various factors during the migrant’s journey on their mental health status and there are no other studies assessing the prevalence of symptoms of depression on Haitian migrants, we cannot generalize the experiences of all Haitians living in Chile as this is a cross-sectional study and includes a small sample. Moreover, our sample size limited the accuracy of our estimates as demonstrated by wide confidence intervals in our analyses. Our study does illustrate an interesting phase of Haitian migrants’ journey that may be useful for the community, health services providers, and future studies conducted in Chile but also throughout the Americas as Haitians are migrating northward [79].

Discussion and conclusion: please consider adding the study´s implications for policy and practice?

We have added a few more sentences in the conclusion that hopefully this study’s findings will provide information for the community, health services providers, and researchers not only in Chile but in other countries in the continent that are receiving Haitian migrants as we speak:

Our study does illustrate an interesting phase of Haitian migrants’ journey that may be useful for the community, health services providers, and future studies conducted in Chile but also throughout the Americas as Haitians are migrating northward [79].

Reviewer 2 Report

In this work, the authors want to investigate depressive symptoms for Haitian migrants living in Chile.

Overall, the article deals with an interesting topic and the authors make it clear why it is important to investigate this social group.

Nonetheless, this study has left me with several perplexities.

FIRST

1) There is no comparison with the inhabitants of Santiago de Chile. Not having a term of comparison, it is not easy to establish what the difference between these two populations is and what it is due to. Indeed, the authors compare their results on depression (22%) with those of the report “Ministerio de Salud Encuesta Nacional de Salud ENS 2009-2010 Chile; 2010 "(17.2%). A difference of only 4.8%, which appears minimal and insufficient to explain the effect of two phenomena: that of Covid-19 (the data on the depression in Chile refer to 2010) and the fact that in general Haitian migrants are particularly prone to depression. This point should be addressed better, perhaps by finding further references in the literature.

2) The COVID-19 issue is really marginal in this study. Apart from the fact that the data was collected during the covid, there are no particular references to it, nor have the authors introduced specific questions to quantify its effect. Similarly to the precedent point, there is no pre- and post-covid comparison in this study. This is particularly limiting, given that the study refers precisely to the effect of COVID-19.

3) In the case of the “Planning to stay in Chile” variable, the two kinds of analysis offer contrasting views. In univariate regression (Table 2), it has an OR <1, so it negatively impacts depression. Conversely, in multivariate regression (Table 3), it has a positive effect on depression (OR> 1). This situation is neither discussed nor justified by the authors. It is essential that the authors justify such a result.

Author Response

Dear Guest Editor

Dr. Ernesto Castañeda and Reviewers,

We appreciate all the comments and suggestions made to our manuscript entitled “Hoping for a better future during COVID-19: How migration plans are protective of depressive symptoms for Haitian migrants living in Chile” (1841747). We have responded to each comment and suggestion and have addressed them in the main document. See our responses below:

Sincerely,

1) There is no comparison with the inhabitants of Santiago de Chile. Not having a term of comparison, it is not easy to establish what the difference between these two populations is and what it is due to. Indeed, the authors compare their results on depression (22%) with those of the report “Ministerio de Salud Encuesta Nacional de Salud ENS 2009-2010 Chile; 2010 "(17.2%). A difference of only 4.8%, which appears minimal and insufficient to explain the effect of two phenomena: that of Covid-19 (the data on the depression in Chile refer to 2010) and the fact that in general Haitian migrants are particularly prone to depression. This point should be addressed better, perhaps by finding further references in the literature.

We have added more data based on other migrant studies and a study that probabilistically sampled households in Santiago de Chile. We hope this addresses the lack of comparison in our discussion and better contextualize our depression prevalence result within the greater Chilean context. Unfortunately, there are very few studies that directly reported on depression prevalence among migrants in Chile, especially in comparison to non-migrants in the same area.

The prevalence of symptoms of depression among a sample of Haitian migrants (n=95) living in Chile was of 22%. This is higher than the prevalence of depression in the general Chilean population – 17.2% [58]. Given that previous research found that migrants had a prevalence of depression between 14% and 20%, our sample presented a higher percentage within the migrant populations [6,59,60]. Furthermore, one cross-sectional study using structured psychiatric interview found that about 5.5% of private households in Santiago de Chile had depression [61].

2) The COVID-19 issue is really marginal in this study. Apart from the fact that the data was collected during the covid, there are no particular references to it, nor have the authors introduced specific questions to quantify its effect. Similarly, to the precedent point, there is no pre- and post-covid comparison in this study. This is particularly limiting, given that the study refers precisely to the effect of COVID-19.

We understand that this may be a concern and we are sorry if we were not clear enough that the pandemic is definitely a critical factor in this paper, not only because the data was collected during the most critical point of the pandemic during a strict lockdown but because there are clear consequences for our participants that we tried to address. In the first place, our literature review illustrates how the pandemic has affected migrants over the world and has had important impacts in their mental health by rising levels of anxiety, depression, and increasing uncertainty overall. Secondly, we had questions in our survey that aimed to capture the consequences of the pandemic on Haitian migrant’s lives, including their mental health status but also their migration plans. In Table 2 we included a few items related to COVID-19 consequences and the participants reported experiencing economic issues (77%), and anxiety and worry (16%). In the same table we show that 84% of the participants who mentioned that had plans to leave Chile said they changed their migration plans due to COVID-19. Therefore, we believe there is a deeper reflection on how the pandemic has affected this particular group and thus we have framed our paper in such complex timing. We have made changes in the discussion so it is clear that we are reflecting not only the time when we conducted the study but also how the pandemic intersects and impacts our participants’ lives and decisions:

As hypothesized, we found that after controlling for relevant covariates, planning to stay in Chile was significantly associated with depressive symptoms. The consequences of the COVID-19 pandemic have been especially harsh on populations that work in informal sector, including migrants [19,20,22], who, given the strict confinement measures implemented in Chile, had very few employment opportunities [56]. An important percentage of this study’s participants (77%) reported experiencing economic issues during the COVID-19 pandemic and when we asked if their migration plans have changed due to the COVID-19 pandemic 84% of those who were planning to leave responded affirmatively. The political climate for migrants in Chile, particularly for Haitian migrants, has been especially difficult as well. The migration reform in 2018 established a consular visa requirement for entry into Chile which had restrictive effects on the arrival of Haitian migrants. In addition, there were administrative obstacles (e.g., the request for a criminal record certificate be sent to Chile from Haiti) which increased monetary costs and waiting times for the completion of regularization procedures and limited the possibilities of regularizing the migratory situation for Haitian migrants already in Chile.

3) In the case of the “Planning to stay in Chile” variable, the two kinds of analysis offer contrasting views. In univariate regression (Table 2), it has an OR <1, so it negatively impacts depression. Conversely, in multivariate regression (Table 3), it has a positive effect on depression (OR> 1). This situation is neither discussed nor justified by the authors. It is essential that the authors justify such a result.

Thank you for noting the inconsistency. We have corrected our variable name in the multivariable regression from “Planning to stay in Chile” to “Planning to leave Chile” to avoid future confusion. We have also reran our regression so that the direction of the odds ratios remain consistent in both the univariate and multivariate regression:

After adjusting for the covariates, planning to leave Chile remained a significant protective  factor for depressive symptoms (Adjusted odds ratio [AOR] = 0.17, 95% CI=0.04 -0.68, p=0.0131).

Round 2

Reviewer 2 Report

I believe the authors have discussed and solved all the issues I raised. The article is now suitable for publication

Author Response

Thank you